Using logical constraints to validate statistical information about disease outbreaks in collaborative knowledge graphs: the case of COVID-19 epidemiology in Wikidata

http://orcid.org/0000-0003-3492-2014 Turki Houcemeddine 1
http://orcid.org/0000-0002-3745-7931 Jemielniak Dariusz 2
http://orcid.org/0000-0002-2786-8913 Hadj Taieb Mohamed A. 1
http://orcid.org/0000-0001-8907-5348 Labra Gayo Jose E. 3
Ben Aouicha Mohamed 1
Banat Mus’ab 4
http://orcid.org/0000-0002-2298-7593 Shafee Thomas 5 6
Prud’hommeaux Eric 7
Lubiana Tiago 8
http://orcid.org/0000-0002-7221-5022 Das Diptanshu 9 10
Mietchen Daniel 11 12 13 14 daniel.mietchen@ronininstitute.org
1 Data Engineering and Semantics Research Unit, Faculty of Sciences of Sfax, University of Sfax , Sfax , Tunisia
2 Department of Management in Networked and Digital Societies, Kozminski University , Warsaw, Masovia , Poland
3 Web Semantics Oviedo (WESO) Research Group, University of Oviedo , Oviedo, Asturias , Spain
4 Faculty of Medicine, Hashemite University , Zarqa , Jordan
5 La Trobe University , Melbourne, Victoria , Australia
6 Swinburne University of Technology , Melbourne, Victoria , Australia
7 World Wide Web Consortium , Cambridge, Massachusetts , United States of America
8 Computational Systems Biology Laboratory, University of São Paulo , São Paulo , Brazil
9 Institute of Child Health (ICH) , Kolkata, West Bengal , India
10 Medica Superspecialty Hospital , Kolkata, West Bengal , India
11 Ronin Institute , Montclair, New Jersey , United States of America
12 Department of Evolutionary and Integrative Ecology, Leibniz Institute of Freshwater Ecology and Inland Fisheries , Berlin , Germany
13 School of Data Science, University of Virginia , Charlottesville, Virginia , United States
14 Institute for Globally Distributed Open Research and Education (IGDORE) , Jena , Germany
Santosh Kc
Electronic publication date: 2022 Sep 29
Publication date: 2022
Volume: 8
Electronic Location ID: e1085
Received 2022 Mar 7; Accepted 2022 Aug 15
Copyright: © 2022 Turki et al.
Copyright year: 2022
Copyright holder: Turki et al.
License: This is an open access article distributed under the terms of the Creative Commons Attribution License, which permits unrestricted use, distribution, reproduction and adaptation in any medium and for any purpose provided that it is properly attributed. For attribution, the original author(s), title, publication source (PeerJ Computer Science) and either DOI or URL of the article must be cited.
License URL: https://creativecommons.org/licenses/by/4.0/

Keywords: SPARQL, Wikidata, Public health surveillance, Knowledge graph refinement, COVID-19 epidemiology, Validation constraints, Data quality, Shape Expressions, Collaborative curation, Public Health Emergency of International Concern

Funding: Ministry of Higher Education and Scientific Research in Tunisia (MoHESR) Wikimedia Foundation WikiCred Grants Initiative of Craig Newmark Philanthropies, Facebook, and Microsoft Spanish Ministry of Economy and Competitiveness TIN2017-88877-R Alfred P. Sloan Foundation G-2019-11458 and G-2021-17106 Polish National Science Center 2019/35/B/HS6/01056 The work done by Houcemeddine Turki, Mohamed Ali Hadj Taieb, and Mohamed Ben Aouicha was supported by the Ministry of Higher Education and Scientific Research in Tunisia (MoHESR) in the framework of Federated Research Project PRFCOV19-D1-P1, by the Wikimedia Foundation through a rapid grant, and by the WikiCred Grants Initiative of Craig Newmark Philanthropies, Facebook, and Microsoft. The work done by Jose Emilio Labra Gayo was funded by the Spanish Ministry of Economy and Competitiveness (Society challenges: TIN2017-88877-R). The work done by Daniel Mietchen was supported by the Alfred P. Sloan Foundation under grant numbers G-2019-11458 and G-2021-17106. The work done by Dariusz Jemielniak was funded by the Polish National Science Center (Grant No. 2019/35/B/HS6/01056). The funders had no role in study design, data collection and analysis, decision to publish, or preparation of the manuscript.

==============================
Urgent global research demands real-time dissemination of precise data. Wikidata, a collaborative and openly licensed knowledge graph available in RDF format, provides an ideal forum for exchanging structured data that can be verified and consolidated using validation schemas and bot edits. In this research article, we catalog an automatable task set necessary to assess and validate the portion of Wikidata relating to the COVID-19 epidemiology. These tasks assess statistical data and are implemented in SPARQL, a query language for semantic databases. We demonstrate the efficiency of our methods for evaluating structured non-relational information on COVID-19 in Wikidata, and its applicability in collaborative ontologies and knowledge graphs more broadly. We show the advantages and limitations of our proposed approach by comparing it to the features of other methods for the validation of linked web data as revealed by previous research.

Introduction

Emerging infectious diseases demand scalable efforts targeting data acquisition, curation, and integration to drive evidence-based medicine, predictive modeling, and public health policy (Dong, Du & Gardner, 2020; Xu, Kraemer & Data Curation Group, 2020). Of particular importance are outbreaks designated as Public Health Emergencies of International Concern, which is currently the case with Polio (Wilder-Smith & Osman, 2020), COVID-19 (Wilder-Smith & Osman, 2020), and Monkeypox (Kozlov, 2022). Building on previous work (cf. Turki et al., 2022) that explored how COVID-19-related information can be collaboratively curated in a knowledge graph, the research presented here zooms in on how statistical information about pathogens, diseases and disease outbreaks can be validated using logical constraints. As before, we will use Wikidata (Vrandečić & Krötzsch, 2014) as an example for such a knowledge graph, while the SARS-CoV-2 virus, the COVID-19 disease as well as the COVID-19 pandemic will serve as examples for curating and validating epidemiological information in such a knowledge graph.

Agile data sharing and computer-supported reasoning about the COVID-19 pandemic and SARS-CoV-2 virus allow us to quickly understand more about the disease’s epidemiology, pathogenesis, and physiopathology. This understanding can then inform the required clinical, scholarly, and public health measures to fight the condition and handle its nonmedical ramifications (Heymann, 2020; Mietchen & Li, 2020; RDA COVID-19 Working Group, 2020). Consequently, initiatives have rapidly emerged to create datasets, web services, and tools to analyze and visualize COVID-19 data. Examples include Johns Hopkins University’s COVID-19 dashboard (Dong, Du & Gardner, 2020) and the Open COVID-19 Data Curation Group’s epidemiological data (Xu, Kraemer & Data Curation Group, 2020). Some of these resources are interactive and return their results based on combined clinical and epidemiological information, scholarly information, and social network analysis (Cuan-Baltazar et al., 2020; Ostaszewski et al., 2020; Kagan, Moran-Gilad & Fire, 2020). However, a significant shortfall in interoperability is common: although these dashboards facilitate examination of their slice of the data, most of them lack general integration with other sites or datasets.

The lack of technical support for interoperability is exacerbated by legal restrictions: despite being free to access, the majority of such dashboards are provided under All Rights Reserved terms or licenses. Similarly, >84% of the 142,665 COVID-19-related projects on the GitHub repository for computing projects are under All Rights Reserved1 terms (as of 4 February 2022). Restrictive licensing of data sets and applications severely impedes their dissemination and integration, ultimately undermining their value for the community of users and re-users. For complex and multifaceted phenomena such as the COVID-19 pandemic, there is a particular need for a collaborative, free, machine-readable, interoperable, and open approach to knowledge graphs that integrate the varied data.

Wikidata (https://www.wikidata.org/) just fits this need as a CC02 licensed, large-scale, multilingual knowledge graph used to represent human knowledge in a structured format (Resource Description Framework or RDF) (Vrandečić & Krötzsch, 2014; Turki et al., 2019). It, therefore, has the advantage of being inherently findable, accessible, interoperable, and reusable, i.e., FAIR (Wilkinson et al., 2016). It was initially developed in 2012 as an adjunct to Wikipedia, but has grown significantly beyond its initial parameters. As of now, it is a centralized, cross-disciplinary meta-database and knowledge base for storing structured information in a format optimized to be easily read and edited by both machines and humans (Erxleben et al., 2014). Thanks to its flexible representation of facts, Wikidata can be automatically enriched using information retrieved from multiple public domain sources or inferred from synthesized data (Turki et al., 2019). This database includes a wide variety of pandemic-related information, including clinical knowledge, epidemiology, biomedical research, software development, geographic, demographic, and genetics data. It can consequently be a vital large-scale reference database to support research and medicine in relation to a pandemic like the still ongoing COVID-19 one (Turki et al., 2019; Waagmeester et al., 2021).

The key hurdle to overcome for projects such as Wikidata is that several of their features can put them at risk of inconsistent structure or coverage: (1) collaborative projects use decentralized contributions rather than central oversight, (2) large-scale projects operate at a scale where manual checking is not possible, and (3) interdisciplinary projects regulate the acquisition of data to integrate a wide variety of data sources. To maximize the usability of the data, it is therefore important to minimize inconsistencies in its structure and coverage. As a result, methods of evaluating the existing knowledge graphs and ontologies, integral to knowledge graph maintenance and development, are of crucial importance. Such an evaluation is particularly relevant in the case of collaborative semantic databases, such as Wikidata.

Knowledge graph evaluation is, therefore, necessary to assess the quality, correctness, or completeness of a given knowledge graph against a set of predetermined criteria (Amith et al., 2018). There are several possible approaches to evaluating a knowledge graph based on external information (so-called extrinsic evaluation), including comparing its structure to a paragon ontology, comparing its coverage to source data, applying it to a test problem and judging the outcomes, and manual expert review of its ontology (Brank, Grobelnik & Mladenic, 2005). Different systematic approaches have been proposed for the comparison of ontologies and knowledge graphs, including NLP techniques, machine learning, association rule mining, and other methods (Lozano-Tello & Gomez-Perez, 2004; Degbelo, 2017; Paulheim, 2017). The criteria for evaluating ontologies typically include accuracy, which determines if definitions, classes, properties, and individual entries in the evaluated ontology are correct; completeness, referring to the scope of coverage of a given knowledge domain in the evaluated ontology; adaptability, determining the range of different anticipated uses of the evaluated ontology (versatility); and clarity, determining the effectiveness of communication of intended meanings of defined terms by the evaluated ontology (Vrandečić, 2009; Obrst et al., 2007; Raad & Cruz, 2015; Amith et al., 2018). However, extrinsic methods are not the only ones that are used for evaluating such a set of criteria. Knowledge graphs can be also assessed through an intrinsic evaluation that assesses the structure of the analyzed knowledge graph thanks to the inference of internal description logics and consistency rules (e.g., Amith et al., 2018).

In the research reported here, we emphasize the use of intrinsic methods to evaluate knowledge graphs by presenting our approach to quality assurance checks and corrections of statistical semantic data in Wikidata, mainly in the context of COVID-19 epidemiological information. This consists of a catalog of automatable tasks based on logical constraints expected of the knowledge graph. Most of these constraints were not explicitly available in the RDF validation resources of Wikidata before the pandemic and are designed in this work to support new types of COVID-19 information in the assessed knowledge graph, particularly epidemiological data. Our approach is built upon the outcomes of previous outbreaks such as the Zika epidemic (Ekins et al., 2016) and aims to pave the way towards handling future outbreaks. We implement these constraints with SPARQL and test them on Wikidata using the public SPARQL endpoint of this knowledge graph, available at https://query.wikidata.org. SPARQL3 is a query language to search, add, modify or delete RDF data available over the Internet without having to retrieve and process the entirety of a given ontological database. We introduce the value of Wikidata as a multipurpose collaborative knowledge graph for the flexible and reliable representation (Wikidata as a Collaborative Knowledge Graph) and validation (Knowledge Graph Validation of Wikidata) of COVID-19 knowledge. Furthermore, we cover the use of SPARQL to query this knowledge graph (Constraint-Driven Heuristics-Based Validation of Epidemiological Data). Then, we demonstrate how statistical constraints can be implemented using SPARQL and applied to verify epidemiological data related to the COVID-19 pandemic (Discussion). Finally, we compare our constraint-based approach with other RDF validation methods through the analysis of the main outcomes of previous research papers related to knowledge graph validation (Conclusion) and conclude future directions.

Wikidata as a collaborative knowledge graph

Wikidata currently serves as a semantic framework for a variety of scientific initiatives ranging from genetics (Burgstaller-Muehlbacher et al., 2016) to invasion biology (Jeschke et al., 2021) and clinical trials (Rasberry et al., 2022), allowing different teams of scholars, volunteers and others to integrate valuable academic data into a collective and standardized pool. Its versatility and interconnectedness are making it an example for interdisciplinary data integration and dissemination across fields as diverse as linguistics, information technology, film studies, and medicine (Turki et al., 2019; Mitraka et al., 2015; Mietchen et al., 2015; Waagmeester, Schriml & Su, 2019; Turki et al., 2017; Wasi, Sachan & Darbari, 2020; Heftberger et al., 2020), including disease outbreaks like those caused by the Zika virus (Ekins et al., 2016) or SARS-CoV-2 (Turki et al., 2022). However, Wikidata’s popularity and recognition across fields still vary significantly (Mora-Cantallops, Sánchez-Alonso & García-Barriocanal, 2019). Its multilingual nature enables fast-updating dynamic data reuse across different language versions of a resource such as Wikipedia (Müller-Birn et al., 2015), with fewer inconsistencies from local culture (Miquel-Ribé & Laniado, 2018) or language biases (Kaffee et al., 2017; Jemielniak & Wilamowski, 2017).

The data structure employed by Wikidata is intended to be highly standardized, whilst maintaining the flexibility to be applied across highly diverse use-cases. There are mainly two essential components: Items, which represent objects, concepts, or topics; and properties, which describe how one item relates to another4 . A statement, therefore, consists of a subject item (S), a property that describes the nature of the statement (P), and an object (O) that can be an item, a value, an external ID, or a string, etc. While items can be freely created, new properties require community discussion and vote, with about 10,000 properties5 currently available. Statements can be further modified by any number of qualifiers to make them more specific, and be supported by references to indicate the source of the information. Thus, Wikidata forms a continuously growing, single, unified network graph, with 99M items forming the nodes, and 1706M statements forming the edges as of July 20, 2022. A live SPARQL endpoint and query service, regular RDF/JSON dumps, as well as linked data APIs and visualization tools, establish a backbone of Wikidata uses (Malyshev et al., 2018; Nielsen, Mietchen & Willighagen, 2017).

Importantly, Wikidata is based on free and open-source philosophy and software and is a database that anyone can edit, similarly to the online encyclopedia, Wikipedia (Jemielniak, 2014). As a result, the emerging ontologies are created entirely collaboratively, without formal pre-publication peer-review (Piscopo & Simperl, 2018), and developed in a community-driven fashion (Samuel, 2017). This approach allows for the dynamic development of areas of interest for the user community but poses challenges, e.g., to systematize and apportion class completeness across topics (Luggen et al., 2019). Also, since the edit history is available to anyone, tracing human and non-human contributions, as well as detecting and reverting vandalism is available by design and relies on community management (Pellissier Tanon & Suchanek, 2019) as well as on software tools like ORES (Sarabadani, Halfaker & Taraborelli, 2017) or the Item Quality Evaluator (https://item-quality-evaluator.toolforge.org/). Wikidata’s quality is overall high, and has been a topic of a number of studies already (e.g., Piscopo & Simperl, 2019; Shenoy et al., 2022).

Other ontological databases and knowledge graphs exist such as DBpedia, Freebase, and OpenCyc (Färber et al., 2018; Pillai, Soon & Haw, 2019). However, much like the factors that led Wikipedia to rise to be a dominant encyclopedia (Shafee et al., 2017; Jemielniak & Wilamowski, 2017), Wikidata’s close connection to Wikimedia volunteer communities and wide readership provided by Wikipedia have quickly given it a competitive edge. The system, therefore, aims to combine the wisdom of the crowds with advanced algorithms. For instance, Wikidata editors are assisted by a property suggesting system, proposing additional properties to be added to entries (Zangerle et al., 2016). Wikidata has subsequently exhibited the highest growth rate of any Wikimedia project and was the first amongst them to pass one billion contributions (Waagmeester et al., 2020).

As a collaborative venture, its governance model is similar to Wikipedia (Lanamäki & Lindman, 2018), but with some important differences. Wide permissions to edit Wikidata are manually granted to approved bots and to Wikimedia accounts that are at least 4 days old and have made at least 50 edits using manual modifications or semi-automated tools for editing Wikidata6 . These accounts are supervised by a limited number of experienced administrators to prevent misleading editing behaviors (such as vandalism, harassment, and abuse) and to ensure a sustainable consistency of the information provided by Wikidata7 . As such, Wikidata is highly relevant to the computer-supported collaborative work (CSCW) field, yet the number of studies of Wikidata from this perspective is still very limited (Sarasua et al., 2019). To understand the value of using SPARQL to validate the usage of relation types in collaborative ontologies and knowledge graphs, it is important to understand the main distinctive features of Wikidata as a collaborative project. Much as Wikidata is developed collaboratively by an international community of editors, it is also designed to be language-neutral. As a result, it is quite possible to contribute to Wikidata with only a limited command of English and to effectively collaborate whilst sharing no common human language—an aspect unique even in the already rich ecosystem of collaborative projects8 (Jemielniak & Przegalinska, 2020). It may well be a cornerstone towards the creation of other language-independent cooperative knowledge creation initiatives, such as Wikifunctions, which is an abstract, language-agnostic Wikipedia currently developed and based on Wikidata (Vrandečić, 2021).

It is also possible to build Wikipedia articles, especially in underrepresented languages, based on Wikidata data only, and create article placeholders to stimulate encyclopedia articles’ growth (Kaffee & Simperl, 2018). This stems from combining concepts that are relatively easily inter-translatable between languages (e.g., professions, causes of death, and capitals) with language-agnostic data (e.g., numbers, geographical coordinates, and dates). As a result, Wikidata is a paragon example of not only cross-cultural cooperation but also human-bot collaborative efforts (Piscopo & Simperl, 2018; Farda-Sarbas et al., 2019). Given the large-scale crowdsourcing efforts in Wikidata and the use of bots and semi-automated tools to mass edit Wikidata, its current volume is higher than what can be reviewed and curated by administrators manually. It is quite intuitive: as the general number of edits created by bots grows, so grows the number of administrative tasks to be automated. Automation may include simplifying alerts, fully and semi-automated reverts, better user tracking, or automated corrections or suggestions. However, the creation of automated methods for the verification and validation of the ontological statements it contains is required most.

Knowledge graph validation of Wikidata

As Wikidata properties are assigned labels, descriptions, and aliases in multiple languages (red in Fig. 2), multilingual information of these properties can be used alongside the labels, descriptions, and aliases of Wikidata items to verify and find sentences supporting biomedical statements in scholarly outputs (Zhang et al., 2019). Such a process can be based on various natural language processing techniques, including word embeddings (Zhang et al., 2019; Chen et al., 2020) and semantic similarity (Ben Aouicha & Hadj Taieb, 2016). These techniques are robust enough to achieve an interesting level of accuracy, and some of them can achieve better accuracy when the Wikidata classes of the subject and object of semantic relations are given as inputs (Lastra-Díaz et al., 2019; Hadj Taieb, Zesch & Ben Aouicha, 2020). The subjects and objects of Wikidata relations can likewise be aligned to other biomedical semantic resources such as MeSH and UMLS Metathesaurus (Turki et al., 2019). Thus, benchmarks for relation extraction based on one of the major biomedical ontologies can be converted into a Wikidata-friendly format9 and used to automatically enrich Wikidata with novel biomedical relations or to automatically find statements supporting existing biomedical Wikidata relations (Zhang et al., 2018). Furthermore, MeSH keywords of scholarly publications can be converted into their Wikidata equivalents, refined using citation and co-citation analysis (Turki, 2018), and used to verify and add biomedical Wikidata relations, e.g., by applying deep learning-based bibliometric-enhanced information retrieval techniques (Mayr et al., 2014; Turki, Hadj Taieb & Ben Aouicha, 2018).

Another option of validating biomedical statements based on the labels and external identifiers of their subjects, predicates, and objects in Wikidata can be the use of these labels and external IDs to find whether the assessed Wikidata statements are available in other knowledge resources (e.g., Disease Ontology) and in open bibliographic databases (e.g., PubMed). Several tools have been successfully built using this principle such as the Wikidata Integrator10 that compares the Wikidata statements of a given gene, protein or cell line with their equivalents in other structured databases like NCBI’s Gene resources, UniProt or Cellosaurus and adjusts them if needed. Complementing this approach, Mismatch Finder (https://www.wikidata.org/wiki/Wikidata:Mismatch_Finder) identifies Wikidata statements that are not available in external databases, while Structured Categories (https://www.wikidata.org/wiki/Wikidata:Structured_Categories) uses SPARQL to identify how the members of a Wikipedia Category are described using Wikidata statements and to reveal whether a statement is missing or mistakenly edited for the definition of category items (Turki, Hadj Taieb & Ben Aouicha, 2021), and RefB11 (Fig. 1) extracts biomedical Wikidata statements not supported by references using SPARQL and identifies the sentences supporting them in scholarly publications using the PubMed Central search engine and a variety of techniques such as concept proximity analysis.

Figure 1 RefB workflow.

Process of RefB, a bot that adds scholarly references to biomedical Wikidata statements based on PubMed Central (Source: https://w.wiki/an$, License: CC BY 4.0). The source code of RefB is available at https://github.com/Data-Engineering-and-Semantics/refb/.

In addition to their multilingual set of labels and descriptions, Wikidata properties are assigned object types using wikibase:propertyType relations (blue in Fig. 2). These relations allow the assignment of appropriate objects to statements, so that non-relational statements cannot have a Wikidata item as an object, while objects of relational statements are not allowed to have data types like a value or a URL (Vrandečić & Krötzsch, 2014).

Figure 2 Example of a Wikidata property and its annotations.

Wikidata page of a clinical property (Source: https://w.wiki/aeF, Derived from: https://w.wiki/aeG, License: CC0). It includes the labels, descriptions, and aliases of the property in multiple languages (red), the object data type (blue), statements where the property is the subject (green) as well as property constraints (brown).

Just like a Wikidata item, a property can be described by statements (green in Fig. 2). The predicates of these statements link a property to its class (instance of (P31)), to its corresponding Wikidata item (subject item of this property (P1629)), to example usages (Wikidata property example (P1855)), to equivalents in other IRIs12 (equivalent property (P1628) and exact match (P2888)), to Wikimedia categories that track its usage on a given wiki (property usage tracking category (P2875)), to its inverse property (inverse property (P1696)), or to its proposal discussion (property proposal discussion (P3254)), etc. These statements can be interesting for various knowledge graph validation purposes. The class, the usage examples, and the proposal discussion of a Wikidata property can be useful through the use of several natural language processing techniques, particularly semantic similarity, to provide several features of the use of the property such as its domain of application (e.g., the subject or object of a statement using a Wikidata property related to medicine should be a medical item) and consequently to eliminate some of the erroneous use by screening the property usage tracking category. The class of the Wikidata item corresponding to the property can be used to identify the field of work of the property and thus flag some inappropriate applications. In addition, the external identifiers of such an item can be used for the verification of biomedical relations by their identification within the semantic annotations of scholarly publications built using the SAT+R (Subject, Action, Target, and Relations) model (Piad-Morffis, Gutiérrez & Muñoz, 2019). The inverse property relations can identify missing statements, which are implied by the presence of inverse statements in Wikidata. However, using inverse properties has the downside that it causes redundancies in the underlying knowledge graph.

Despite the importance of these statements defining properties, property constraint (P2302) relations (brown in Fig. 2) are the semantic relations that are primarily used for the validation of the usage of a property. In essence, they define a set of conditions for the use of a property, including several heuristics for the type and format of the subject or the object, information about the characteristics of the property, and several description logics for the usage of the property as shown in Table 1. Property constraints are either manually added by Wikidata users or inferred with high accuracy from the knowledge graph of Wikidata or the history of human changes to Wikidata statements (Pellissier Tanon, Bourgaux & Suchanek, 2019; Hanika, Marx & Stumme, 2019).

Table 1 Constraint types for the usage of Wikidata properties.

Each property constraint is given with its Wikidata identifier, an English label and an English description.

Wikidata ID	Constraint type	Description	
Q19474404	Single value constraint	Constraint used to specify that this property generally contains a single value per item	
Q21502404	Format constraint	Constraint used to specify that the value for this property has to correspond to a given pattern	
Q21502408	Mandatory constraint	Status of a Wikidata property constraint: indicates that the specified constraint applies to the subject property without exception and must not be violated	
Q21502410	Distinct values constraint	Constraint used to specify that the value for this property is likely to be different from all other items	
Q21510852	Commons link constraint	Constraint used to specify that the value must link to an existing Wikimedia Commons page	
Q21510854	Difference within range constraint	Constraint used to specify that the value of a given statement should only differ in the given way. Use with qualifiers minimum quantity/maximum quantity	
Q21510856	Mandatory qualifier constraint	Constraint used to specify that the listed qualifier has to be used	
Q21510862	Symmetric constraint	Constraint used to specify that the referenced entity should also link back to this entity	
Q21510863	Used as qualifier constraint	Constraint used to specify that a property must only be used as a qualifier	
Q21510864	Value requires statement constraint	Constraint used to specify that the referenced item should have a statement with a given property	
Q21510495	Relation of type constraint	Relation establishing dependency between types/meta-levels of its members	
Q21510851	Allowed qualifiers constraint	Constraint used to specify that only the listed qualifiers should be used. Novalue disallows any qualifier	
Q21510865	Value type constraint	Constraint used to specify that the referenced item should be a subclass or instance of a given type	
Q21514353	Allowed units constraint	Constraint used to specify that only listed units may be used	
Q21510857	Multi-value constraint	Constraint used to specify that a property generally contains more than one value per item	
Q21510859	One-of constraint	Constraint used to specify that the value for this property has to be one of a given set of items	
Q21510860	Range constraint	Constraint used to specify that the value must be between two given values	
Q21528958	Used for values only constraint	Constraint used to specify that a property can only be used as a property for values, not as a qualifier or reference	
Q21528959	Used as reference constraint	Constraint used to specify that a property must only be used in references or instances of citation (Q1713)	
Q25796498	Contemporary constraint	Constraint used to specify that the subject and the object have to coincide or coexist at some point in history	
Q21502838	Conflicts-with constraint	Constraint used to specify that an item must not have a given statement	
Q21503247	Item requires statement constraint	Constraint used to specify that an item with this statement should also have another given property	
Q21503250	Type constraint	Constraint used to specify that the item described by such properties should be a subclass or instance of a given type	
Q54554025	Citation needed constraint	Constraint specifies that a property must have at least one reference	
Q62026391	Suggestion constraint	Status of a Wikidata property constraint: indicates that the specified constraint merely suggests additional improvements, and violations are not as severe as for regular or mandatory constraints	
Q64006792	Lexeme value requires lexical category constraint	Constraint used to specify that the referenced lexeme should have a given lexical category	
Q42750658	Value constraint	Class of constraints on the value of a statement with a given property. For constraint: use specific items (e.g., “value type constraint”, “value requires statement constraint”, “format constraint”, etc.)	
Q51723761	No bounds constraint	Constraint specifies that a property must only have values that do not have bounds	
Q52004125	Allowed entity types constraint	Constraint used to specify that only listed entity types are valid for this property	
Q52060874	Single best value constraint	Constraint used to specify that this property generally contains a single “best” value per item, though other values may be included as long as the “best” value is marked with a preferred rank	
Q52558054	None of constraint	Constraint specifying values that should not be used for the given property	
Q52712340	One-of qualifier value property constraint	Constraint used to specify which values can be used for a given qualifier when used on a specific property	
Q52848401	Integer constraint	Constraint used when values have to be integer only	
Q53869507	Property scope constraint	Constraint to define the scope of the property (main value, qualifier, references, or combination); only supported by KrBot currently	

As shown in Fig. 2 a property constraint is defined as a relation where the property type is featured as an object, and the detailed conditions of the constraint to be applied on Wikidata statements are integrated as qualifiers to the relation. When a statement uses a property in a way that does not conform to its corresponding constraint, the statement is automatically included in the property constraint report (https://www.wikidata.org/wiki/Wikidata:Database_reports/Constraint_violations) and is marked by an exclamation mark on the page of the subject item (Fig. 3), so that either the item can be repaired by the community or by Wikidata bots, or the property constraint can be renegotiated.

Figure 3 Example of a property constraint violation indicated via the Wikidata user interface.

On the page of the Wikidata item Q3603152 (flash blindness), a constraint violation is indicated by the encircled exclamation mark. Clicking on it reveals the display of the popup with some further explanation (File available on Wikimedia Commons: https://w.wiki/ZuJ, License: CC0).

Although these methods are important to verify and validate Wikidata, they are not the only ones that are used for these purposes. Various MediaWiki templates, Lua modules or bots can be used to check, flag and in some cases fix inconsistencies. For instance, the Autofix template (https://www.wikidata.org/wiki/Template:Autofix) allows to specify regex patterns that then trigger bot edits, e.g., to enforce case normalization of values for a given property.

In 2019, Wikidata announced the adoption of the Shape Expressions language (ShEx) as part of the MediaWiki entity schemas extension (https://www.mediawiki.org/wiki/Extension:EntitySchema). ShEx was proposed following an RDF validation workshop that was organized by W3C (https://www.w3.org/2012/12/rdf-val/report) in 2014 as a concise, high-level language to describe and validate RDF data (Prud’hommeaux, Labra Gayo & Solbrig, 2014). This Mediawiki extension uses ShEx to store structure definitions (EntitySchemas or Shapes) for sets of Wikidata entities that are selected by some query pattern (frequently the involvement of said entities in a Wikidata class). This provides collaborative quality control where the community can iteratively develop a schema and refine the data to conform to that schema. For those familiar with XML, ShEx is analogous to XML Schema or RelaxNG. SHACL (Shapes Constraint Language), another language used to constraint RDF data models, uses a flat list of constraints, analogous to XML’s Schematron. SHACL was adapted from SPIN (SPARQL Inference Notation) by the W3C Data Shapes working group in 2014 and became a W3C recommendation in 2017 (Knublauch & Kontokostas, 2017). However, ShEx was chosen to represent EntitySchemas in Wikidata, as it has a compact syntax that makes it more human-friendly, supports recursion, and is designed to support distributed networks of reusable schemas (Labra Gayo et al., 2017). Besides the possibility to infer ShEx expressions from the screening of a large set of concerned items, they can be intuitively written by humans.

In Wikidata, ShEx-based EntitySchemas are assigned an identifier (a number beginning with an E) as well as labels, descriptions, and aliases in multiple languages, so that they can be identified by users. Entity schemas are defined using the ShEx-compact syntax13 , which is a concise, human-readable syntax. A schema usually begins with some prefix declarations similar to those in SPARQL. An optional start definition declares the shape which will be used by default. In the example (Fig. 4), the shape <app> will be used, and its declaration contains a list of properties, possible values, and cardinalities. By default, shapes are open, which means that other properties apart from the ones declared are allowed. In this example, the values of property wdt:P31 are declared to be either a COVID-19 dashboard (wd:Q90790055), a search engine (wd:Q91136116), or a dataset (wd:Q91137337). The EXTRA directive indicates that there can be additional values for property wdt:P31 that differ from the specified ones. The value for property wdt:P1476 is declared to be zero or more literals. The cardinality indicators come from regular expressions, where ‘?’ means zero or one, ‘*’; means zero or more, and ‘+’ means one or more. While the values for the other properties are declared to be anything (the dot indicates no constraint) zero or more times, except for the properties wdt:P577 and wdt:P7103 that are marked as optional using the question mark. Further documentation about ShEx can be found at http://shex.io/ and in Labra Gayo et al. (2017).

Figure 4 Entity Schema example.

Entity Schema for COVID-19 dashboards, search engines and datasets (Source: https://www.wikidata.org/wiki/EntitySchema:E205. File available on Wikimedia Commons: https://w.wiki/4rg5, License: CC0).

Due to the ease of using ShEx to define EntitySchemas, it has been used successfully to validate Danish lexemes in Wikidata (Nielsen, Thornton & Labra-Gayo, 2019) and biomedical Wikidata statements (Thornton et al., 2019). During the COVID-19 pandemic, Wikidata’s data model of every COVID-19-related class as well as of all major biomedical classes has been converted to an EntitySchema, so that it can be used to validate the representation of COVID-19 Wikidata statements (Waagmeester et al., 2021). These EntitySchemas were successfully used to enhance the development and the robustness of the semantic structure of the data model underlying the COVID-19 knowledge graph in Wikidata and are accordingly made available at a subpage of Wikidata’s WikiProject COVID-1914 . Significant efforts are currently underway to further simplify the definition of EntitySchemas by making them more intuitive and concise, enabling an increase of the usage of ShEx to validate semantic knowledge in Wikidata (Samuel, 2021).

Beyond these interesting methods, validation constraints can be inferred and used to verify semantic statements in a knowledge graph through the use of the full screening of RDF dumps or the use of SPARQL queries. RDF dumps are particularly used for screening Wikidata items in a class to identify common features of the assessed entities based on a set of formal rules (Marx & Krötzsch, 2017; Hanika, Marx & Stumme, 2019). These features involve common characteristics of the data model of the concerned class with patterns of used Wikidata properties such as symmetry and are later used to verify the completeness of the class and validate the statements related to the evaluated class. The analysis of RDF dumps for Wikidata can be coupled with the federated screening of the RDF dumps of other knowledge graphs such as DBpedia, allowing to evaluate Wikidata shapes based on aligned external structures for the same domain (Ahmadi & Papotti, 2021). Nowadays, efforts are provided to extend inference-based methods for the validation of Wikidata through the development of probabilistic approaches to identify when a statement is unlikely to be defined for an item allowing to enhance the evaluation of the completeness of Wikidata as an open knowledge graph (Arnaout et al., 2021). As SPARQL has been designed to extract a searched pattern from a semantic graph (Pérez, Arenas & Gutierrez, 2009), it has been used to query and harmonize competency questions15 , and to evaluate ontologies and knowledge graphs in a context-sensitive way (Vasanthapriyan, Tian & Xiang, 2017; Bansal & Chawla, 2016; Martin, 2018). Indeed, a sister project presents how SPARQL can be used to generate data visualizations16 (Nielsen, Mietchen & Willighagen, 2017; Shorland, Mietchen & Willighagen, 2020). Validating RDF data portals using SPARQL queries has been regularly proposed as an approach that gives great flexibility and expressiveness (Labra Gayo & Alvarez Rodríguez, 2013). However, academic literature is still far from revealing a consensus on methods and approaches to evaluate ontologies using this query language (Walisadeera, Ginige & Wikramanayake, 2016), and other approaches have been proposed for validation (Thornton et al., 2019; Labra-Gayo et al., 2019). Currently, there is mostly an effort to normalize how to define SPARQL queries, particularly for knowledge graph validation purposes, to save runtime and ameliorate the completeness of the output of a query using a set of heuristics and axioms (Salas & Hogan, 2022).

In Wikidata, the Wikidata Query Service (https://query.wikidata.org) allows querying the knowledge graph using SPARQL (Malyshev et al., 2018; Turki et al., 2019). The query service includes a specific endpoint (https://query.wikidata.org/sparql) that allows programmatic access to SPARQL queries in programming languages. The required Wikidata prefixes are already supported in the backend of the service and do not need to be defined (Malyshev et al., 2018). What the user needs to do is to formulate their SPARQL query (black in Fig. 5) and click on the Run button (blue in Fig. 5). After an execution period of up to 60 s, the results will appear (green in Fig. 5) and can be downloaded in different formats (brown in Fig. 5), including JSON, TSV, CSV, HTML, and SVG. Different modes for the visualization of the query results can be chosen (purple in Fig. 5), particularly table, charts (line, scatter, area, bubble), image grid, map, tree, timeline, and graph. The query service also allows users to use a query helper (red in Fig. 5) that can generate basic SPARQL queries, and to get inspired by sample queries (yellow in Fig. 5), especially when they lack experience. It also allows users to generate a short link for the query (pink in Fig. 5) and code snippets to embed the query results in web pages and computer programs (brown in Fig. 5) (Malyshev et al., 2018).

Figure 5 Web interface of the Wikidata Query Service.

It involves a query field (black), a query builder (red), a short link button (pink), a Run button (blue), a visualization mode button (purple), a download button (brown), an embedding code generation button (grey), a results field (green), and a sample query button (yellow) (Source: https://w.wiki/aeH, Derived from: https://query.wikidata.org, License: CC0).

Constraint-driven heuristics-based validation of epidemiological data

The characterization of epidemiological data is possible using a variety of statistical measures that show the acuteness, the dynamics, and the prognosis of a given disease outbreak. These measures include the cumulative count17 of cases (P1603 (199569 statements; orange in Fig. 6), noted c, as defined before), deaths (P1120 (243250 statements18 ; black in Fig. 6), noted d), recoveries (P8010 (36119 statements; green in Fig. 6), noted r), clinical tests (P8011 (21249 statements; blue in Fig. 6), noted t), and hospitalized cases (P8049 (5755 statements; grey in Fig. 6), noted h) as well as several measurements done by the synthesis of the values of epidemiological counts such as case fatality rate (P3457 (51504 statements; red in Fig. 6), noted m), basic reproduction number (P3492, noted R0), minimal incubation period in humans (P3488, noted mn), and maximal incubation period in humans (P3487, noted mx) (Rothman, Greenland & Lash, 2008). For all these statistical data, every information should be coupled by a point in time (P585, noted Z) qualifier defining the date of the stated measurement and by a Determination method (P459, noted Q) qualifier identifying the measurement method of the given information as these variables are subject to change over days or according to used methods of computation.

Figure 6 Sample statistical data available through Wikidata.

The item about the COVID-19 pandemic in Tunisia is shown (Adapted from: https://www.wikidata.org/wiki/Q87343682, Source: https://w.wiki/uUr, License: CC0).

From count statistics (c, t, d, h, and r statements), it is possible to compare regional epidemiological variables and their variance for a given date (Z) or date range, and relate these to the general disease outbreak (each component defined as a part of (P361) of the general outbreak) as shown in Table 2. Such comparisons are enabled using statistical conditions that are commonly used in epidemiology (Zu et al., 2020). Tasks V1 and V2 have been generated from the evidence that COVID-19 started in late 2019 and that its clinical discovery can only be done through medical diagnosis techniques (Zu et al., 2020). Tasks V3 and V4 have been derived from the fact that c, d, r, and t are cumulative counts. Consequently, these variables are only subjects to remain constant or increase over days. Task V5 is motivated by the fact that an epidemiological count cannot return negative values. Tasks V6, V7, V8, and V9 are due to the evidence that d, r, and h cannot be superior to c as COVID-19 deaths are the consequence of severe infections by SARS-CoV-2 that can only be managed in hospitals (Rothman, Greenland & Lash, 2008) and as a patient needs to undergo COVID-19 testing to be confirmed as a case of the disease (Zu et al., 2020). V10 is built upon the assumption that c, d, r, h, and t values can be geographically aggregated (Rothman, Greenland & Lash, 2008).

Table 2 Tasks for the heuristics-based evaluation of epidemiological data using the Wikidata SPARQL endpoint.

Each validation task is given with its identifier, a brief description of the heuristic validation criteria and an example where the data does not fit them. See the section “Constraint-driven heuristics-based validation of epidemiological data” for definitions of the epidemiological variables.

Task	Description	Sample filtered deficient statement	
Validating qualifiers of COVID-19 epidemiological statements	
V1	Verify Z as a date > November 01, 2019	COVID-19 pandemic in X <number of cases> 5 <point in time> March 25, 20	
V2	Verify Q as any subclass of (P279*) of medical diagnosis (Q177719)	COVID-19 pandemic in X <number of cases> 5 <point in time> March 25, 2020 <determination method> COVID-19 Dashboard	
Ensuring the cumulative pattern of c, d, r, and t	
V3	Identify c, d, r and t statements having a value in date Z+1 not superior or equal to the one in date Z (Verify if dZ ≤ dZ+1, rZ ≤ rZ+1, tZ ≤ tZ+1, and cZ ≤ cZ+1)	(COVID-19 pandemic in X <number of cases> 5 <point in time> March 25, 2020) AND (COVID-19 pandemic in X <number of cases> 6 <point in time> March 24, 2020)	
V4	Find missing values of c, d, r and t in date Z+1 where corresponding values in dates Z and Z+2 are equal	(COVID-19 pandemic in X <number of cases> 5 <point in time> March 24, 2020) AND (COVID-19 pandemic in X <number of cases> 6 <point in time> March 26, 2020) AND (COVID-19 pandemic in X <number of cases> no value <point in time> March 25, 2020)	
Validating values of epidemiological data for a given date	
V5	Identifying c, d, r, h, and t statements with negative values	COVID-19 pandemic in X <number of cases> -5 <point in time> March 25, 2020	
V6	Identify h statements having a value superior to the number of cases for a date Z	(COVID-19 pandemic in X <number of hospitalized cases> 15 <point in time> March 25, 2020) AND (COVID-19 pandemic in X <number of cases> 5 <point in time> March 25, 2020)	
V7	Identify c statements having a value superior or equal to the number of clinical tests for a date Z	(COVID-19 pandemic in X <number of clinical tests> 4 <point in time> March 25, 2020) AND (COVID-19 pandemic in X <number of cases> 5 <point in time> March 25, 2020)	
V8	Identify c statements having a value inferior to the number of deaths for a date Z	(COVID-19 pandemic in X <number of deaths> 10 <point in time> March 25, 2020) AND (COVID-19 pandemic in X <number of cases> 5 <point in time> March 25, 2020)	
V9	Identify c statements having a value inferior to the number of recoveries for a date Z	(COVID-19 pandemic in X <number of recoveries> 10 <point in time> March 25, 2020) AND (COVID-19 pandemic in X <number of cases> 5 <point in time> March 25, 2020)	
V10	Comparing the epidemiological variables of a general outbreak with the ones of its components	(COVID-19 pandemic in X <number of cases> 10 <point in time> March 25, 2020) AND (COVID-19 pandemic in Y <number of cases> 5 <point in time> March 25, 2020) WHERE X is a district of Y	

This task set has been applied using ten simple SPARQL queries that can be found in Appendix A where <PropertyID> is the Wikidata property to be analyzed and has returned 5,496 inconsistencies in the COVID-19 epidemiological information (as of August 8, 2020) as shown in Table 3. Among these potentially inaccurate statements, 2,856 were number of cases statements, 2,467 were number of deaths statements, 189 were number of recoveries statements, nine were number of clinical tests statements, and 10 were number of hospitalized cases statements. This distribution of the inconsistencies among epidemiological properties is explained by the dominance of number of cases and number of deaths statements on the COVID-19 epidemiological information. Most of these inconsistencies are linked to a violation of the cumulative pattern of major variables. These issues can be resolved using tools for the automatic enrichment of Wikidata like QuickStatements (cf. Turki et al., 2019) or adjusted one by one by active members of WikiProject COVID-19.

Table 3 Matrix overview of data quality issues identified per validation task and epidemiological Wikidata property.

Rows represent validation tasks as defined in Table 2, columns the corresponding epidemiological Wikidata properties, and the value in a given cell represents the number of deficient statements identified by the row’s specific task for the column’s epidemiological Wikidata property on a given date (August 8, 2020).

	c	d	r	t	h	Overall	
V1	18	9	10	2	1	40	
V2	2	91	6	0	0	99	
V3	660	92	6	5		763	
V4	2,081	2,247	149	1		4,478	
V5	0	0	0	0	0	0	
V6	8				8	8	
V7	1			1		1	
V8	9	9				9	
V9	17		17			17	
V10	60	19	1	0	1	81	
Overall	2,856	2,467	189	9	10	5,496	

Concerning the variables issued from the integration of basic epidemiological counts (m, R0, mn, and mx statements), they give a summary overview of the statistical behavior of the studied infectious pandemic and that is why they can be useful to identify whether the stated evolution of the morbidity and mortality caused by the outbreak is reasonable (Delamater et al., 2019). However, the validation of these variables is more complicated due to the complexity of their definition (Delamater et al., 2019; Backer, Klinkenberg & Wallinga, 2020; Li et al., 2020). The basic reproduction number (R0) is meant to be a constant that characterizes the dissemination power of an infectious agent. It is defined as the expected number of people (within a community with no prior exposure to the disease) that can contract a disease via the same infected individual. This variable should exceed the threshold of one to define a contagious disease (Delamater et al., 2019). Although R0 can give an idea about the general behavior of an outbreak of a given disease, any calculated value depends on the model used for its computation (e.g., SIR Model) as well as the underlying data and is consequently a bit imprecise and variable from one study to another (Delamater et al., 2019). That is why it is not reliable to use this variable to evaluate the accuracy of epidemiological counts for a given pandemic. The only heuristic that can be applied to this variable is to verify if its value exceeds one for diseases causing large outbreaks. The incubation period of a disease gives an overview of the silent time required by an infectious agent to become active in the host organism and cause notable symptoms (Backer, Klinkenberg & Wallinga, 2020; Li et al., 2020). This variable is very important, as it reveals how many days an inactive case can spread the disease in the host’s environment before the host is being symptomatically identified. As a result, it can give an idea about the contagiousness of the infectious disease and its basic reproduction number (R0). However, the determination of the incubation period—especially for a novel pathogen—is challenging, as a patient often cannot identify with precision the day when they had been exposed to the disease, at least if they did not travel to an endemic region or had not been in contact with a person they knew to be infected. This factor was behind the measurement of falsely small incubation periods for COVID-19 at the beginning of the COVID-19 epidemic in China (Backer, Klinkenberg & Wallinga, 2020). Furthermore, the use of minimal (mn) and maximal (mx) incubation periods in Wikidata to epidemiologically describe a disease instead of the median incubation period is a source of a lack of accuracy of the extracted values (Backer, Klinkenberg & Wallinga, 2020; Li et al., 2020). Minimal and maximal incubation periods for a given disease are obtained in the function of the mean ( −X) and standard deviation ( σ) of the measures of the confidence interval of observed incubation periods in patients. Effectively, mn is equal to −X−z∗σn and mx is equal to −X+z∗σnw where n is the number of analyzed observations and z is a characteristic of the hypothetical statistical distribution and of the statistical confidence level adopted for the estimation (Altman et al., 2013). As a consequence, mn and mx variables are modified according to the number of observations (n) with a smaller difference between the two variables for higher values of n. The two measures also vary according to the used statistical distribution and that is why different values of mn and mx were reported for COVID-19 when applying different distributions (Weibull, gamma, and log-normal distribution) using a confidence level of 0.95 on the same set of observed cases (Backer, Klinkenberg & Wallinga, 2020). Similarly, the two variables can change according to the adopted confidence level (p − 1) when using the same statistical distribution where a higher confidence level is correlated with a higher difference between the calculated mn and mx values, as shown in Fig. 7 (Ward & Murray-Ward, 1999; Altman et al., 2013). Given these reasons and despite the significant importance of the two measures, these two statistical variables cannot be used to evaluate statistical epidemiological counts for COVID-19 due to their lack of precision and difficulty of determination.

Figure 7 Distribution statistics.

Confidence intervals for different p-values (p) when using a normal distribution (Source: https://w.wiki/aKT, License: Public Domain) (after Ward & Murray-Ward, 1999).

As for the reported case fatality rate (m), it is the quotient of the cumulative number of deaths (d) and the cumulative number of cases (c) as stated in official reports. It is consequently straightforward to validate for a given disease by comparing its values with reported counts of cases and deaths (Rothman, Greenland & Lash, 2008). Here, two simple heuristics can be applied using SPARQL queries as shown in Appendix B. As the number of deaths is less than or equal to the number of cases of a given disease, m values should be set between 0 and 1. That is why Task M1 is defined to extract m statements where m > 1 or m < 0. Also, as m = d/c for a date Z, m values that are not close to the corresponding quotients of deaths by disease cases should be identified as deficient and m values should be stated for a given date Z if mortality and morbidity counts exist. Thus, Task M2 is created to extract m values where the absolute value of (m − d/c) is superior to 0.001, and Task M3 is developed to identify (item, date) pairs where m statements are missing and c and d statements are available in Wikidata. Absolute values for Task M2 are obtained using SPARQL’s ABS function, and deficient (item, date) pairs are eliminated in Task M3 where m > 1 and c < d.

As a result of these three tasks, we interestingly identified 143 problematic m statements and 7,116 missing m statements. 133 of the problematic statements are identified thanks to Task M2 and concern 25 Wikidata items and 31 distinct dates, and only 10 deficient statements related to three Wikidata items and eight distinct dates are found using Task M1. These statements should be checked against reference datasets to verify their values and to determine the reason behind their deficiency. Such a reason can be the integration of the wrong case and death counts in Wikidata, or a bug or inaccuracy within the source code of the bot making or updating such statements. The verification process can be automatically done using an algorithm that compares Wikidata values (c, d, and m statements) with their corresponding ones in other databases (using file or API reading libraries) and subsequently adjusts statements using the Wikidata API directly or via tools like QuickStatements (Turki et al., 2019). As for the missing m statements returned by M3, they are linked to 395 disease outbreak items and to 205 distinct dates and concern 70% (7116/10168) of the (case count, death count) pairs available in Wikidata. The outcome of M3 proves the efficiency of comparative constraints to enrich and assess the completeness of epidemiological data available in a knowledge graph, particularly Wikidata, based on existing information. Consequently, derivatives of Task M3 can build to infer d values based on c and m statements or to find c values based on d and m statements. The missing statements found by such tasks can be integrated in Wikidata using a bot based on Wikidata API and Wikidata Query Service to ameliorate the completeness and integrity of available mortality data for epidemics, mainly the COVID-19 pandemic (Turki et al., 2019).

Discussion

The results presented here demonstrate the value of our statistical constraints-based validation approach for knowledge graphs like Wikidata across a range of features (Tables 2 and 3). These tasks successfully address most of the competency questions, particularly conceptual orientation (clarity), coherence (consistency), strength (precision), and full coverage (completeness). Combined with previous findings in the context of bioinformatics (Bolleman et al., 2020; Marx & Krötzsch, 2017; Darari et al., 2020), this proves that the efficiency of rule-based approaches to evaluate semantic information from scratch displays a similar accuracy as other available ontology evaluation algorithms (Amith et al., 2019; Zhang & Bodenreider, 2010). The efficiency of these constraint-based assessment methods can be further enhanced by using machine learning techniques to perform imputations and adjustments on deficient data (Bischof et al., 2018). The scope of rule-based methods can be similarly expanded to cover other competency questions such as non-redundancy (conciseness) through the proposal of other logical constraints to tackle them, such as a condition to find taxonomic relations to trim in a knowledge graph (examples can be found at https://www.wikidata.org/wiki/Wikidata:Database_evaluation). The main limitation of applying the logical constraints using SPARQL in the context of Wikidata is that the runtime of a query that infers or verifies a complex condition or that analyzes a huge amount of class items or property use cases can exceed the timeout limit of the used endpoint (Malyshev et al., 2018; Chah & Andritsos, 2021). Here, the inference of logical constraints and the identification of inconsistent semantic information through the analysis of full dumps of Wikidata can be more efficient, although this comes with advanced storage and processing requirements (Chah & Andritsos, 2021). Another option can be either the loading of a Wikidata dump and the running of those queries on a designated SPARQL endpoint with more permissive timeout settings or the use of one of the publicly available clones, though their data is usually less complete19 .

These evaluation assignments covered by our approach can be done by other rule-based (structure-based and semantic-based) ontology evaluation methods. Structure-based methods verify whether a knowledge graph is defined according to a set of formatting constraints, and semantic-based methods check whether concepts and statements of a knowledge graph meet logical conditions (Amith et al., 2018). Some of these methods are software tools, particularly Protégé extensions such as OWLET (Lampoltshammer & Heistracher, 2014) and OntoCheck (Schober et al., 2012). OWLET infers the JSON schema logics of a given knowledge graph, converts them into OWL-DL axioms, and uses the semantic rules to validate the assessed ontological data (Lampoltshammer & Heistracher, 2014). OntoCheck screens an ontology to identify structural conventions and constraints for the definition of the analyzed relational information and consequently to homogenize the data structure and quality of the ontology by eliminating typos and pattern violations (Schober et al., 2012). Here, the advantage of applying constraints using SPARQL is that its runtime is faster, as it does not require the download of the full dumps of the evaluated knowledge graph (Malyshev et al., 2018). The benefit of our method and other structure-based and semantic-based web-based tools for knowledge graph validation like OntoKeeper (Amith et al., 2019) and adviseEditor (Geller et al., 2013), when compared to software tools, is that the maximal size of the knowledge graphs that can be assessed by web services is larger than the one that can be evaluated by software tools because the latter depends on the requirements and capacities of the host computer (Lampoltshammer & Heistracher, 2014; Schober et al., 2012). These drawbacks of other structure-based tools can indeed be solved through the simplification of the knowledge graph by reducing redundancies using techniques like ontology trimming (Jantzen et al., 2011) or through the construction of an abstraction network to decrease the complexity of the analyzed knowledge graph (Amith et al., 2018; Halper et al., 2015). However, knowledge graph simplification processes are time-consuming, and resulting time gain can consequently be insignificant (Jantzen et al., 2011; Amith et al., 2018; Halper et al., 2015).

Such tasks can also be solved using data-driven ontology evaluation methods. These techniques process texts in natural languages to validate the concepts and statements of a knowledge graph and currently include intrinsic (lexical-based) and extrinsic (cross-validation, big data-based, and corpus-based) methods (Amith et al., 2018). Lexical-based methods use rules implemented in SQL or SPARQL to retrieve items and glosses corresponding to a concept and their semantic relations (mostly subclass of statements) (Rector & Iannone, 2012; Luo, Mejino & Zhang, 2013). These items are then compared against a second set of rules to identify inconsistencies in their labels, descriptions, or semantic relations (Amith et al., 2018). The output can then be analyzed using natural language processing techniques such as hamming distance measures (Luo, Mejino & Zhang, 2013), semantic annotation tools (Rector & Iannone, 2012), and semantic similarity measures (Amith et al., 2018) to comparatively identify deficiencies in the semantic representation, labelling, and symmetry of the assessed knowledge graph. Conversely, extrinsic data-based methods extract the usage and linguistic patterns from raw text corpuses such as bibliographic databases and clinical records (Corpus-based methods) or from gold standard semantic resources like large ontologies and knowledge graphs (Cross-validation methods) or social media posts and interactions, Internet of Things data or web service statistics (Big data-based methods) (Amith et al., 2018; Sebei, Hadj Taieb & Aouicha, 2018; Rector, Brandt & Schneider, 2011; Gangemi et al., 2005) using structure-based and semantic-based ontology evaluation methods as explained above (Rector, Brandt & Schneider, 2011) as well as a range of techniques including machine learning (Bean et al., 2017; Zhang et al., 2018), topic modeling using Latent Dirichlet Analysis (Abd-Alrazaq et al., 2020), word embeddings (Zhang et al., 2019), statistical correlations (Vanderkam et al., 2013) and semantic annotation methods (Li et al., 2016). The returned features of the analyzed resources are compared to the ones of the analyzed knowledge graph to assess the accuracy and completeness of the definition and use of concepts and properties (Amith et al., 2018).

When compared to our proposed approach, lexical-based methods have the advantage to identify and adjust characteristics of a knowledge graph item based on its natural language information of a knowledge graph item, particularly terms and glosses (Rector & Iannone, 2012; Luo, Mejino & Zhang, 2013). The drawback of using semantic similarity, word embeddings, and topic modeling techniques in such approaches is that these techniques are sensitive to the used parameters, to input characteristics, and to the chosen models of computation and can consequently give different results according to the context of determination (Lastra-Díaz et al., 2019; Hadj Taieb, Zesch & Ben Aouicha, 2020). The current role of constraints in the extraction of lexical information and respective semantic relations (Rector & Iannone, 2012; Luo, Mejino & Zhang, 2013) proves that the scope of constraint-based validation should not only be restricted to rule-based evaluation but also to lexical-based evaluation. Yet, the function of logical conditions should be expanded to refine the list of pairs (lexical information, semantic relation) to more accurately identify deficient and missing semantic relations and defective lexical data and to support multilingual lexical-based methods. This would build on the many SPARQL functions that analyze strings in knowledge graphs20 such as STRLEN (length of a string), STRSTARTS (verification of a substring beginning a given string), STRENDS (verification of a substring finishing a given string), and CONTAINS (verification of a substring included in a given string) (DuCharme, 2013; Harris, Seaborne & Prud’hommeaux, 2013).

As for the extrinsic data-driven methods, they are mainly based on large-scale resources that are regularly curated and enriched. Raw-text corpora are mainly composed of scholarly publications (Raad & Cruz, 2015) and blog posts (Park et al., 2016). Information in scholarly publications is ever-changing according to the dynamic advances in scholarly knowledge, particularly medical data (Jalalifard, Norouzi & Isfandyari-Moghaddam, 2013). This expansion of scientific information in scholarly publications is highly recognized in the context of COVID-19 where detailed information about COVID-19 disease and the SARS-CoV-2 virus is published within less than 6 months (Kagan, Moran-Gilad & Fire, 2020). Big data is the set of real-time statistical and textual information that is generated by web services including search engines and social media and by the Internet of Things objects including sensors (Sebei, Hadj Taieb & Aouicha, 2018). This data is characterized by its value, variety, variability, velocity, veracity, and volume (Sebei, Hadj Taieb & Aouicha, 2018) and can be consequently used to track the changes of the community knowledge and consciousness over time (Abd-Alrazaq et al., 2020; Turki et al., 2020). Large semantic resources are ontologies and knowledge graphs that are built and curated by a community of specialists and that are regularly verified, updated, and enriched using human efforts and computer programs (Lee et al., 2013). These resources represent broad and reliable information about a given specialty through machine learning techniques (Zhang et al., 2018) and the crowdsourcing of scientific efforts (Mortensen et al., 2014) and can be consequently compared to other semantic databases for validation purposes. Examples of these resources are the COVID-19 Disease Map (Ostaszewski et al., 2020) and SNOMED-CT21 (Lee et al., 2013).

Large-scale knowledge graphs are dynamic corpora. Changes in the logical and semantic conditions for the definition of knowledge in a particular domain need to be identified to adjust the assessed knowledge graph accordingly. Rule-based and lexical-based approaches (especially constraints-based methods) are therefore less simple to apply than extrinsic data-driven methods (Amith et al., 2018). Nonetheless, the growing and changing nature of gold-standard resources require continuous human efforts and an advanced software architecture to maintain (e.g., structure-based and semantic-based methods), process (e.g., word embeddings and latent Dirichlet analysis), and store (e.g., Hadoop and MapReduce) these reference resources (Mortensen et al., 2014; Lee et al., 2013; Sebei, Hadj Taieb & Aouicha, 2018). This architecture has advanced hardware requirements and its results are subject to change according to the used parameters (Sebei, Hadj Taieb & Aouicha, 2018).

These tasks are in line with the usage of Shape Expressions as well as property constraints and relations for the validation of data quality and completeness of the semantic information of class items in knowledge graphs as shown in the “Knowledge graph validation of Wikidata” section. A ShEx ShapeMap is a pair of a triple pattern for selecting entities to validate and a shape against which to validate them. This allows for the definition of the properties to be used for the items of a given class (Prud’hommeaux, Labra Gayo & Solbrig, 2014; Waagmeester et al., 2021) and property constraints and relations based on the meta-ontology (i.e., data skeleton) of Wikidata. Expressions written in shape-based property usage validation languages for RDF (e.g., SHACL) can be used to state conditions and formatting restrictions for the usage of relational and non-relational properties (Erxleben et al., 2014; Thornton et al., 2019; Gangemi et al., 2005). SPARQL can be more efficient in inferring such information than the currently existing techniques that screen all the items and statements of a knowledge graph one by one to identify the conditions for the usage of properties (e.g., SQID) mainly because SPARQL is meant to directly extract information according to a pattern without having to evaluate all the conditions against all items of a knowledge graph (Marx & Krötzsch, 2017; Hanika, Marx & Stumme, 2019; Pérez, Arenas & Gutierrez, 2009).

The separate execution of value-based constraints is common in the quality control of XML data. Typically, structural constraints are managed by RelaxNG or XML Schemas, while value-based constraints are captured as Schematron. Much as Schematron rules are typically embedded in RelaxNG, the consistency constraints presented above can be embedded in Shape Expressions Semantic Actions or in SHACL-SPARQL as shown in Fig. 8 (Melo & Paulheim, 2020). These supplement structural schema languages with mechanisms to capture value-based constraints and in doing so, provide context for the enforcement of those constraints. The implementation of value-based constraints shown in the “Constraint-driven heuristics-based validation of epidemiological data” section can likewise be implemented in a shapes language (Labra-Gayo et al., 2019). Parsing the rules in Table 2 would allow the mechanical generation or augmentation of shapes, providing flexibility for how the rules are expressed while still exploiting the power of shape languages for validation. More generally, ontology-based and knowledge graph-based software tools have the potential to provide wide data and platform interoperability, and thus their semantic interoperability is relevant for a range of downstream applications such as IoT and WoT technologies (Gyrard, Datta & Bonnet, 2018).

Figure 8 Key elements of data quality workflows on Wikidata.

Interactions between consistency rules, property statements, and RDF validation languages (Source: https://w.wiki/ao5, License: CC BY 4.0).

Conclusion

In this article, we investigate how to best assess epidemiological knowledge in collaborative ontologies and knowledge graphs using statistical constraints which we describe based on the example of COVID-19 data in Wikidata. Collaborative databases produced through the cumulative edits of thousands of users can generate huge amounts of structured information (Turki et al., 2019) but as a result of their rather uncoordinated development, they often lead to uneven coverage of crucial information and inconsistent expression of that information. The resulting gaps are a significant problem (conflicting values, reasoning deficiencies, and missing statements). Avoiding, identifying, and closing these gaps is therefore of top importance. We presented a standardized methodology for auditing key aspects of data quality and completeness for these resources22 .

This approach complements and informs shape-based methods for data conformance to community-decided schemas. The SPARQL execution does not require any pre-processing, and is not only applicable to the validation of the representation of a given item according to a reference data model but also to the comparison of the assessed statistical statements. Our method is demonstrated as useful for measuring the overall accuracy and data quality on a subset of Wikidata and thus highlights a necessary first step in any pipeline for detecting and fixing issues in collaborative ontologies and knowledge graphs.

This work has shown the state of the knowledge graph as a snapshot in time. Future work will extend this to investigate how the knowledge base evolves as more biomedical knowledge is integrated into it over time. This will require incorporating the edit history in the SPARQL endpoint APIs of knowledge graphs (Pellissier Tanon & Suchanek, 2019; Dos Reis et al., 2014) to dynamically visualize time-resolved SPARQL queries. We will also couple the information inferred using this method23 with Shape Expressions and the explicit constraints of relation types to provide a more effective enrichment, refinement, and adjustment of collaborative ontologies and knowledge graphs with statistical data. This will be an excellent infrastructure to enable the support of non-relational information. Although our article focuses on COVID-19, applying the basic approach outlined here to any disease outbreak - especially those designated as Public Health Emergencies of International Concern24 , which have particularly urgent needs for efficient sharing and quality assessment of data—would require minimal modification, since Wikidata is progressing quickly towards greater integration with biomedical reference data. Consequently, frameworks similar to the one presented here can be designed for validating other types of biomedical data as well as data from other knowledge domains. We look forward to extending our proposed approach to allow knowledge graphs to handle non-relational statements about future epidemics and other disasters such as earthquakes as well as to clinical trials.

Supplemental Information

Supplemental Information 1 SPARQL queries for the heuristics-based validation of epidemiological counts in Wikidata.

The SPARQL queries that were used for the Tasks defined in Table 2, to be run against the Wikidata Query Service available at https://query.wikidata.org/. Note that this query service has Wikidata-specific prefixes predefined, so they do not need to be re-stated in a query.

Click here for additional data file.

Supplemental Information 2 SPARQL queries for the validation of case fatality rate statements in Wikidata.

These SPARQL queries correspond to the Tasks M1, M2 and M3 that address heuristics concerning the case fatality rate m.

Click here for additional data file.

We thank the Wikidata community, Olivier Corby (Université Côte d’Azur, France), Odile Papini (Aix-Marseille Université, France), Egon Willighagen (Maastricht University, Netherlands), and Mahir Morshed (University of Illinois at Urbana-Champaign, United States of America) for useful comments and discussions about the topic of this research paper. This research paper is published on behalf of the WikiProject COVID-19 members: Jan Ainali, Susanna Ånäs, Erica Azzellini, Mus’ab Banat, Mohamed Ben Aouicha, Alessandra Boccone, Jane Darnell, Diptanshu Das, Lena Denis, Rich Farmbrough, Daniel Fernández-Álvarez, Konrad Foerstner, Jose Emilio Labra Gayo, Mauricio V. Genta, Mohamed Ali Hadj Taieb, James Hare, Alejandro González Hevia, David Hicks, Toby Hudson, Netha Hussain, Jinoy Tom Jacob, Dariusz Jemielniak, Krupal Kasyap, Will Kent, Samuel Klein, Jasper J. Koehorst, Martina Kutmon, Antoine Logean, Tiago Lubiana, Andy Mabbett, Kimberli Mäkäräinen, Tania Maio, Bodhisattwa Mandal, Nandhini Meenakshi, Daniel Mietchen, Nandana Mihindukulasooriya, Mahir Morshed, Peter Murray-Rust, Minh Nguyễn, Finn Årup Nielsen, Mike Nolan, Shay Nowick, Julian Leonardo Paez, João Alexandre Peschanski, Alexander Pico, Lane Rasberry, Mairelys Lemus-Rojas, Diego Saez-Trumper, Magnus Sälgö, John Samuel, Peter J. Schaap, Jodi Schneider, Thomas Shafee, Nick Sheppard, Adam Shorland, Ranjith Siji, Michal Josef Špaček, Ralf Stephan, Andrew I. Su, Hilary Thorsen, Houcemeddine Turki, Lisa M. Verhagen, Denny Vrandečić, Andra Waagmeester, and Egon Willighagen.

Additional Information and Declarations

Competing Interests

Author Contributions

Data Availability

1 120,109 of 142,665 as of 4 February 2022: https://github.com/search?q=covid-19+OR+covid19+OR+coronavirus+OR+cord19+OR+cord-19

2 CC0 is a rights waiver similar to Creative Commons licenses, used to publish material into the public domain. It waives as much copyright as possible within a given jurisdiction. Further information can be found at https://creativecommons.org/publicdomain/zero/1.0/.

3 An open license SPARQL textbook available in multiple languages can be found at https://en.wikibooks.org/wiki/SPARQL.

4 Detailed information about the data structure of Wikidata can be found in Turki et al. (2022).

5 For an updated list of available Wikidata properties, please see https://tools.wmflabs.org/hay/propbrowse/.

6 For an overview of the semi-automated editing tools for Wikidata, please see https://www.wikidata.org/wiki/Wikidata:Tools.

7 Further information about the rights and governance of users in Wikidata is shown at https://www.wikidata.org/wiki/Wikidata:User_access_levels.

8 For further details about the language representation of COVID-19 knowledge in Wikidata, please refer to Turki et al. (2022), which has a figure and multiple tables on the subject.

9 A Wikidata-friendly format of a database is an edition of that resource where items and predicates of triples are replaced by their equivalents in Wikidata or in ontologies integrated with it.

10 Wikidata Integrator is a bot framework for automatically curating genetic information provided by Wikidata (https://github.com/SuLab/WikidataIntegrator). For Wikidata bots using this framework, refer to https://www.wikidata.org/wiki/Wikidata:WikiProject_Gene_Wiki#Bot_accounts. The framework has been adapted to various specific contexts, e.g., the curation of cell lines indexed in Cellosaurus, as per https://github.com/calipho-sib/cellosaurus-wikidata-bot.

11 RefB: Description at https://www.wikidata.org/wiki/Wikidata:Requests_for_permissions/Bot/RefB_(WikiCred), Source code at https://github.com/Data-Engineering-and-Semantics/refb/, Wikidata edits at https://www.wikidata.org/wiki/Special:Contributions/RefB_(WikiCred).

12 Internationalized Resource Identifier (IRI) is a standardized character string (e.g., a URL) that recognizes a given item in a semantic resource

13 ShEx schemas can also be defined in RDF-based representations like Turtle or JSON-LD.

14 The data models for WikiProject COVID-19 are accessible via https://www.wikidata.org/wiki/Wikidata:WikiProject_COVID-19/Data_models.

15 Competency questions: A set of requirements ensuring consistency of a knowledge graph, constraints determining what knowledge to be involved in a knowledge graph (Wiśniewski et al., 2019).

16 For SPARQL-based visualizations of COVID-19 information in Wikidata, see https://speed.ieee.tn/, https://egonw.github.io/SARS-CoV-2-Queries/, https://www.wikidata.org/wiki/Wikidata:WikiProject_COVID-19/Queries, and https://scholia.toolforge.org/topic/Q84263196.

17 We found the Wikidata properties reflecting epidemiological data about COVID-19 outbreaks using a specific SPARQL query available at https://w.wiki/5UsE. Please note that current results can return new properties that did not exist as of August 8, 2020 such as Number of vaccinations (P9107).

18 As of August 8, 2020. For updated statistics, see https://w.wiki/Z5m.

19 For instance, the query SELECT (COUNT(*) AS ?c) WHERE {?s ?p ?o} currently gives 11857528152 results on the clone at https://wikidata.demo.openlinksw.com/sparql that was set up by Chalupsky et al. (2021), while the live Wikidata result as of 23 July 2022 is 14040950269.

20 Detailed information about string functions in SPARQL can be found at https://www.w3.org/TR/sparql11-query/#func-strings.

21 Systematized Nomenclature Of Medicine—Clinical Terms

22 This method can be adapted to meet the needs of the user. For instance, the SPARQL queries can be slightly adjusted to assess other patterns in collaborative ontologies such as the usage of classes.

23 This information can be represented in the form of RDF triples where the subject is the studied relation type and integrated into Wikidata.

24 Epidemiological data about the monkeypox epidemic have begun to be tracked, e.g. via the item Q112070734 for the 2022 monkeypox outbreak and similar entries with a more regional focus like Q112059351 for the 2022 monkeypox outbreak in the United Kingdom.

All the co-authors of this paper except Eric Prud’hommeaux are active members of WikiProject Medicine, the community curating clinical knowledge in Wikidata, and of WikiProject COVID-19, the community developing multidisciplinary COVID-19 information in Wikidata. Dariusz Jemielniak is a non-paid voluntary member of the Board of Trustees of the Wikimedia Foundation, the non-profit publisher of Wikipedia and Wikidata. Eric Prud’hommeaux is a co-creator of SPARQL. Eric Prud’hommeaux and Jose E Labra Gayo are co-creators of ShEx.

Houcemeddine Turki conceived and designed the experiments, performed the experiments, analyzed the data, performed the computation work, prepared figures and/or tables, authored or reviewed drafts of the article, and approved the final draft.

Dariusz Jemielniak conceived and designed the experiments, authored or reviewed drafts of the article, and approved the final draft.

Mohamed A. Hadj Taieb conceived and designed the experiments, authored or reviewed drafts of the article, and approved the final draft.

Jose E. Labra Gayo conceived and designed the experiments, performed the experiments, analyzed the data, performed the computation work, authored or reviewed drafts of the article, and approved the final draft.

Mohamed Ben Aouicha conceived and designed the experiments, authored or reviewed drafts of the article, and approved the final draft.

Mus’ab Banat conceived and designed the experiments, performed the experiments, authored or reviewed drafts of the article, and approved the final draft.

Thomas Shafee conceived and designed the experiments, performed the experiments, analyzed the data, performed the computation work, prepared figures and/or tables, authored or reviewed drafts of the article, and approved the final draft.

Eric Prud’hommeaux conceived and designed the experiments, performed the experiments, analyzed the data, performed the computation work, authored or reviewed drafts of the article, and approved the final draft.

Tiago Lubiana conceived and designed the experiments, performed the experiments, analyzed the data, performed the computation work, authored or reviewed drafts of the article, and approved the final draft.

Diptanshu Das conceived and designed the experiments, authored or reviewed drafts of the article, and approved the final draft.

Daniel Mietchen conceived and designed the experiments, performed the experiments, analyzed the data, performed the computation work, prepared figures and/or tables, authored or reviewed drafts of the article, and approved the final draft.

The following information was supplied regarding data availability:

All the SPARQL queries used in this research work are available in the Appendices.

The Internet Archive URLs cited in this article are available at Wikidata: https://web.archive.org/save/https://www.wikidata.org/w/index.php?title=User:Daniel_Mietchen/sandbox&oldid=1580603965.

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
