# Peer review of "Using logical constraints to validate statistical information about disease outbreaks in collaborative knowledge graphs: the case of COVID-19 epidemiology in Wikidata"

_PeerJ Computer Science, doi:10.7717/peerj-cs.1085_

## Round 0.1 · original submission · Minor Revisions

Point-by-point responses are required.

Reviewer 1 ·

Basic reporting

The paper provides an excellent review of Wikidata and existing methods for evaluation. Then it details a method for validating numerical data in the collaborative knowledge graph, using known epidemiological constraints expressed as SPARQL queries. This is a general technique that can be used for data in other domains, which can be a very useful addition to the ontologist toolbox.

Experimental design

The problem is always well stated and the relationship between the new technique and existing methods is clearly elaborated.
There is plenty of detail to show how the method is implemented and the results are clear.

Validity of the findings

No comment.

Additional comments

This is a very nicely written paper and I can see myself using the method in my own work. It is very nice to see work which is scientifically sensible as well as useful in practice.

·

Basic reporting

The manuscript is easily readable. However, there are some minor inconsistencies in the manuscript that need updating. I have attached the PDF I used for this review with specific remarks posted as added notes.

1. On line 148 the authors describe the process of adding content to Wikidata as an upload process. This is inaccurate, by design, wikidata does not host primary data. The applicable notability constraints require data transformation when data is aligned with wikidata.
2. Line 170: Wikidata indeed delivers RDF dumps, but in two variants, the complete dumps and the "truthy" dump. The primary dump, however, is available as a JSON dump. This distinction does need to be made. (https://www.wikidata.org/wiki/Wikidata:Database_download)
3. Line 176: This is inaccurate. The property instantiation in Wikidata is done through a proposal phase where an admin (centralized rules) decides when a property is fit for purpose. This fact constitutes some level of centralization. I would reword this by calling the process peer-review.
4. Line: 251: This is inaccurate. The Wikidata Integrator does not take Wikidata as input, but rather takes a primary source as input and then compares those items to Wikidata to align that source with Wikidata. If a match is found (by matching references and statements) the item is updated, if not a new item is added to align and represent that primary resource in Wikidata. Unmatched items on Wikidata are left alone.
5. Line 274: There are more equivalent properties in Wikidata such as P2888 which aligns using SKOS.
Furthermore, there are normalized properties wdtn: and pn:, which makes properties of type external identifier that contain a statement with property "formatter URI for RDF resource (P1921)" linking out to external linked data resources.
6. Line 291: The inverse properties are redundant since they can also be identified by using SPARQL. Using inverse properties has the downside that it causes redundancies in the underlying knowledge graph.
7. Line 305: Property constraints are suggestions for improvement, and as such are not necessarily a violation. Since Wikidata combines many resources there will be cases of disagreement between resources. E.g. if we have two databases addressing the same topic while having different levels of specialization. When the curator of the least specialized database dictates a constraint, the statement addressing the items from the more specialized database will lead to a multitude of constraint reports. This is why the constraints shown do not necessarily mean that the statement needs to be fixed, it might also mean that the constraints written are inaccurate.
8. Line 373: In theory this is true, but the RDF dumps are currently often too big that substantial infrastructure is necessary to load and process the RDF dumps of Wikidata alone. I am not aware of efforts where that loading is extended with DBpedia. The cited reference to support this claim is behind a paywall, so I could not verify the claim.
9. Line 561: Why not load a Wikidata clone and run those queries on a designated sparql endpoint without limits, or use one of the publically available clones e.g. https://wikidata.demo.openlinksw.com/sparql

Minor notes:
* line: 128 is Zika a pandemic or an endemic? The supporting citation is behind a paywall, so I could not verify that claim.
* line: 149: "its versatility and interconnections are making it a standard". Standard implies a specification, I would reword this into "making is an example".
* line 163-168: Maybe add a figure like https://www.researchgate.net/figure/Wikidata-item-and-data-organization-Wikidata-items-can-be-added-or-edited-by-anyone_fig4_298795180 to illustrate the anatomy of a wikidata item.
* line 169: Please add a date stamp
* line 175: Remove "very popular" it does not add anything to the report. Or list how popular it is.
* line 206: Please provide a figure showing that the labels are indeed equally distributed across the language. e.g. Does all properties have an equal number of translations?
* line 240: What is a wikidata friendly format?
* line 395: Please consider adding a description about query.wikidata.org/sparql which allows programmatic access to SPARQL queries in programming languages. The WDQS also provides snippets in various programming languages that support integration in automatic pipelines
* Line 414 Please don't use "simple". I would recommend replacing the word simple with "straightforward". Yes for someone who went through the learning curve, those applications might seem simple, however for readers who do not know about the infrastructure, things might look not simple. In one case, however, I would actually not use neither "simple" nor "straightforward".
* Line 516: The queries shown in Appendix B are not standard SPARQL, but specific to the underlying (blazegraph) endpoint. A disclaimer should be made that these queries are specific to blazegraph.
* Line 737: The link does not resolve

Experimental design

no comment

Validity of the findings

In general, the findings seem valid. However, the queries used in Appendix A & B, do not exclude deprecated values. This might distort the reported numbers. I would recommend adding a filter for deprecated-ranked statements to verify those do not exist in the numbers reported based on the queries in Appendix B.

Additional comments

Although COVID19 is indeed the subject in this paper, however, the general application of what is reported is broader than the COVID19 use case, which might also vouch for a more generic title.

·

Basic reporting

The paper is generally well written, although there are minor issues with the English throughout.

The topic of assessing data quality is clearly presented at a general level. The authors are missing discussions around the quality of Wikidata itself

Piscopo, A. and Simperl, E., 2019, August. What we talk about when we talk about Wikidata quality: a literature survey. In *Proceedings of the 15th International Symposium on Open Collaboration* (pp. 1-11).

Shenoy, K., Ilievski, F., Garijo, D., Schwabe, D. and Szekely, P., 2022. A Study of the Quality of Wikidata. *Journal of Web Semantics*, *72*, p.100679.

The reference to the FAIR Data Principles on line 80 should be updated the original paper on FAIR data since that is what the authors are referring to. The current citation does not seem to cover the discussion.

Wilkinson, M.D., Dumontier, M., Aalbersberg, I.J., Appleton, G., Axton, M., Baak, A., Blomberg, N., Boiten, J.W., da Silva Santos, L.B., Bourne, P.E. and Bouwman, J., 2016. The FAIR Guiding Principles for scientific data management and stewardship. *Scientific data*, *3*(1), pp.1-9.

The structure of the paper is fine but the narrative within the paper could be improved. The sections up to line 410 give a very verbose and sometimes rambling introduction to Wikidata. The Discussion section is equally unfocused to the work at hand. These should both be condensed and given more focus to the topics required for understanding the content of the work. At present the paper is very imbalanced with the actual work accounting for only a small part of the paper.

Figures, tables, and raw data are generally fine. I would expect English statements to describe the queries in the appendices to make them self-contained. Are these queries also available for execution through the wikidata query service; if so a link would be really helpful? The grey cells in Table 2 should be explained.

Experimental design

The paper presents an approach of using ShEx and SPARQL queries to assess the quality of epidemiological data related to COVID-19 stored in Wikidata. One ShEx expression is provided together with 10 SPARQL queries used to identify data quality issues. An aggregation of running the SPARQL queries is presented within the paper.

Validity of the findings

The approach is not particularly novel, but the work is sound and provides a useful mechanism to assess the quality of Wikidata data. This is done for a point in time but it is unclear how much data this is conducted over. How much data about COVID-19 was present in Wikidata as of 8 August 2020? How many of the SPARQL queries could be executed over the public endpoint? How have the authors verified that their queries have covered all COVID-19 epidemiological data; were subtypes considered?

While the focus on COVID-19 is very understandable at the current time, it seems to me that the approach would be applicable to any disease. Perhaps the authors could add a discussion about the more general applicability of the approach.

Additional comments

Line 147: More than one initiative should be cited to substanititate the claim that Wikidata is used for a variety of scientific initiatives. On the same line, it is not just scholars who can update Wikidata.

Line 185: The names of databases should be provided

Line 399: compilation should be replace with execution

---

## Round 0.2 · accepted · Accept

Sufficient revisions have been made. The revised manuscript can be accepted.